# Patients’ Health Experiences of Post COVID-19 Condition—A Qualitative Study

**DOI:** 10.3390/ijerph192113980

**Published:** 2022-10-27

**Authors:** Johanna Almgren, Emma Löfström, Julia S Malmborg, Jens Nygren, Johan Undén, Ingrid Larsson

**Affiliations:** 1School of Health and Welfare, Halmstad University, P.O. Box 823, SE-301 18 Halmstad, Sweden; 2Department of Clinical Microbiology, Hallands Hospital Halmstad, SE-301 85 Halmstad, Sweden; 3Department of Clinical Sciences, Lund University, P.O. Box 117, SE-221 00 Lund, Sweden; 4Department of Operation and Intensive Care, Hallands Hospital Halmstad, SE-301 85 Halmstad, Sweden; 5Spenshult Research and Development Centre, SE-302 74 Halmstad, Sweden; 6Section of Rheumatology, Department of Clinical Sciences, Lund University, P.O. Box 117, SE-221 00 Lund, Sweden

**Keywords:** COVID-19, experience, health, lifestyle, long-term symptoms

## Abstract

Patients who suffer from long-term symptoms of COVID-19, described as post COVID-19 condition, are a new and large group of patients. There is a lack of knowledge concerning health experiences in this patient group. The aim of this study was to explore patients’ health experiences of post COVID-19 condition. Data collection was performed through 14 semi-structured interviews. The qualitative content analysis resulted in six sub-categories, three categories, and an overall theme. Patients experienced symptoms of varying duration—ranging from 5–21 months. The results showed that patients’ health experiences of post COVID-19 condition moved between uncertainty and new insights. This was shown by patients experiencing: (1) loss of abilities, including losing smell and taste and lacking energy, (2) loss of control, including being foreign to oneself and seeking answers, and (3) revaluation of life, including accepting the transformed body and prioritizing health. This study illustrates that patients living with post COVID-19 condition need to be treated with flexibility, based on each individual’s unique challenges and experiences of the symptoms and the consequences for their health.

## 1. Introduction

COVID-19, a disease caused by severe acute respiratory syndrome coronavirus 2 (SARS-CoV-2), was first discovered at the end of 2019 and has since caused a worldwide pandemic that, at time of writing, is ongoing. Common symptoms of COVID-19 are fever, cough, and sore throat, but there is a wide range—from asymptomatic cases to severe cases with respiratory failure and sometimes fatal outcomes [1]. In the majority of patients, the symptoms are mild and transient, although it has become evident that some patients experience prolonged symptoms [2,3]. Even patients with mild initial symptoms can experience long-term symptoms or post COVID-19 condition [4,5]. According to the definition presented by the World Health Organization, post COVID-19 condition occurs in individuals with a history of probable or confirmed SARS-CoV-2 infection, usually 3 months from the onset of COVID-19 with symptoms and that last for at least 2 months and cannot be explained by an alternative diagnosis [6]. The frequency of post COVID-19 condition varies in different studies but has been observed to affect 11.7% up to 61.9% of the patients with COVID-19 [7,8,9]. Common symptoms of post COVID-19 condition are dyspnea, fatigue, and cognitive dysfunction [3,9,10]. The precise effect of COVID-19 on the health of the population is difficult to determine, but research indicates that the disease has negatively impacted public health [11,12,13,14]. Studies show that, for some specific patients, COVID-19 has caused a negative impact on their health and lifestyle and, thereby, their lives. It happens that patients experience post-traumatic stress after COVID-19, which may be due to, for example, isolation from loved ones [15,16].

COVID-19 has brought about a change in the lifestyle among patients. The changes are due to physical distancing and self-isolation, which have strongly affected eating habits and everyday behaviour, in particular [12]. Another factor is that COVID-19 has burdened healthcare services, which has meant that other physical and mental diseases have not been investigated and have sometimes been left untreated [11]. Patients who have survived an acute phase of COVID-19 are at increased risk of mental illness. Symptoms can include anxiety, depression, stress, impaired neurocognitive functioning, and sleep disorders [14].

In post COVID-19 condition, symptoms such as headache, fatigue, sleep disturbances, anxiety, cough, and/or breathlessness are common [3]. The prevalence of different symptoms within the population suffering from post COVID-19 condition are not often specified, but fatigue is in some studies reported to affect between 58 up to 74% [3,4], headache is reported in some studies in 44 up to 71% [3,17] and dizziness up to 59% [17]. Sometimes the symptoms are more diffuse and can also fluctuate or relapse over time. Some patients experience difficulty concentrating and fatigue with minimal physical exertion, affecting work and everyday life [16,18,19]. Other symptoms that can occur in post COVID-19 condition are smell and taste disorders, commonly loss of smell/taste or altered smell sensations, which indicates that the nervous system is affected [20]. It has also been shown that COVID-19 can lead to cognitive impairment in several areas, which can persist for several months [4,21]. Hartung et al. [22] describes cognitive impairment as a common symptom of post COVID-19 condition, and Hellgren et al. [4] reports cognitive impairment in 29% of the patients.

Patients who suffer from post COVID-19 condition are a new and large group of patients [23]. Experiencing COVID-19 in itself can be stressful and pose a threat to health, both mentally and physically [3,24], and there is a lack of knowledge concerning the consequences of the post COVID-19 condition. There is a need to increase our understanding of post COVID-19 condition to increase effect of treatment and rehabilitation. Therefore, this study aimed to explore patients’ health experiences of post COVID-19 condition.

## 2. Materials and Methods

### 2.1. Design

The study had an explorative design based on qualitative content analysis, with an inductive approach. Qualitative content analysis aims to systematically analyze qualitative data to provide knowledge shaped by an interaction between the researcher and the participant. The analysis focuses on identifying similarities and differences in the texts’ content, describing variations, and increasing understanding of the participants’ experiences of, for example, health [25]. The study complies with the consolidated criteria for reporting qualitative research (COREQ) [26].

### 2.2. Patients

The inclusion criterion for participation in the study was that the patients had Polymerase Chain Reaction (PCR)-confirmed COVID-19 with persistent symptoms for at least two months after subsidence. The patients are included in an ongoing study on COVID-19, “COVID-19 in Region Halland: Symptoms and immunity”—a study with 360 non-hospitalized patients, divided into one prospectively included cohort (*n* = 154) and one retrospectively included cohort. The patients are followed with a weekly digital health questionnaire and repeated blood samples for immunological tests [27]. From the weekly questionnaires in the prospective cohort, 18 patients with symptoms lasting more than eight weeks could be identified. The reported symptoms fluctuated over time, but fatigue and anosmia/taste disorder were dominating (Table 1). All patients (*n* = 18) were asked to participate in this interview study in a written informed consent letter and 14 chose to participate. The patients had been diagnosed with COVID-19 between June and December 2020, and 12 of them experienced continued symptoms at the time of the interviews. The patients varied in age from 26 to 76, with a median age of 58. Ten women and four men participated, and all 14 were born in Sweden (Table 1). Of the four eligible patients who did not participate, one declined, one could not be reached, and two gave verbal consent, but did not send signed consent, despite having received reminders.

### 2.3. Data Collection

Data were collected through individual semi-structured interviews and conducted by the last author (IL), an experienced qualitative researcher. The interviews began with the researcher clarifying the aim of the study. A semi-structured interview guide was used, with open-ended questions focused on the patients’ experiences of health during and after COVID-19. The main questions were “How do you describe your experiences of being affected by COVID-19?”, “What does health mean for you?”, and “How do you experience your health after the disease?”. The patients were encouraged to provide more in-depth information, by asking them to “Tell us more” or with questions such as “What do you mean?” Two pilot interviews were conducted to test the interview questions. The pilot interviews lasted between 92 to 101 min. As no amendment was required, these interviews were included in the study. The interviews were conducted by telephone (*n* = 10) or through a digital forum, Microsoft Teams^®^ (*n* = 4), and lasted for 45–120 min. The total time for the interviews was 17 h and 55 min. The interviews were audio-recorded and transcribed verbatim.

### 2.4. Data Analysis

Data were analysed with qualitative content analysis [25,28]. The interviews were listened to, and the transcriptions were read several times carefully, as a first step. The purpose of reading the data material several times was to get acquainted with the content and create a sense of the whole. In the second step, content that was related to the aim of the study was extracted from the transcriptions. This extracted content formed the so-called unit of analysis. The units of analysis were then divided into smaller meaning units (*n* = 221), which were condensed and labelled with a code. The codes were then interpreted according to their differences and similarities, which resulted in six sub-categories. For example, the codes “gratitude for one’s health” and “reflection on the fragility of life”, formed the sub-category “prioritizing health”. In the next step, the sub-categories were grouped into three categories, which reflected the manifest content, i.e., what the text said. The latent content, the underlying meaning, was formulated and described as an overarching theme. Data were coded and manually analyzed without Software by the first author (J.A.) in dialogue with co-authors (I.L. and J.S.M.), and throughout the process there was a continuous shift between analyzing the whole text and parts of the text. The analysis process was discussed until a consensus was reached in the research group. The intention was to stay close to the text and preserve contextual meaning.

### 2.5. Ethical Considerations

The study was approved by the Swedish Ethical Review Authority (no. 2020-02691, 2022-00339-02). The study complies with the ethical principles set out for research on humans by the Declaration of Helsinki [29]. The Declaration of Helsinki [29] states the importance of the researcher protecting and being responsible for the health, integrity, dignity, right to self-determination, and confidentiality of the participants. All patients were given oral and written information about the aim of the study. Participation was voluntary, and patients gave written informed consent to participate. The patients were allowed to choose a time for the interview, and they also received oral information that personal data, audio recordings, and the transcribed interviews would be treated confidentially, in accordance with guidelines presented by the Swedish Research Council [30]. The patients in this study have been protected and their well-being has been considered. No risks of participation have been identified, and the material was available only to the researchers. The patients could withdraw at any time during the interview without giving any reason and were offered the opportunity to share and discuss thoughts and feelings that might arise [29,30].

## 3. Results

Patients’ health experiences of post COVID-19 condition moved between uncertainty and new insights. This was described as: (1) loss of abilities, including losing smell and taste and lacking energy, (2) loss of control, including being foreign to oneself and seeking answers, and (3) revaluation of life, including accepting the transformed body and prioritizing health (Table 2). Loss of abilities and loss of control contributed to uncertainty and challenges for the patients. Besides experiencing ill health, patients also experienced a revaluation of life which contributed to new insights regarding priorities, family life, and health.

### 3.1. Loss of Abilities

Post COVID-19 condition was experienced as loss of abilities, such as losing smell and taste and lacking energy. Loss of abilities constituted a new, foreign feeling that sometimes completely took one over. For example, the fatigue that characterized everyday life was difficult to deal with. Eventually, the loss of abilities took on a deeper meaning and affected the whole of the person’s life.

#### 3.1.1. Losing Smell and Taste

Patients described that, before suffering from the COVID-19 infection, smell and taste were taken for granted. Only after their loss was it understood how important these senses were for the experience of health. Health was affected, for example, physically through changes in diet, and mentally around worries about the loss. The patients experienced that loss of smell and taste negatively impacted other aspects of their health and their lifestyle. The desire to cook could decrease, which in turn could lead to a poorer diet. Previously appreciated activities were negatively affected by the loss of smell and taste; for example, going for walks by the sea and grilling sausages while not sensing the smell or the taste. The lack of smell and taste experience led to a reduced number of activities than before the COVID-19 disease, which also applied to social contexts.


*It’s no fun to invite people home and cook, you spend a lot of time around food. You grill or go out and eat. To go out and eat at a restaurant is to throw the money into the lake right now. Sure, the experience, but you do not sense any aromas in the restaurant either. Often it can smell like cigarette smoke even though it is not cigarette smoke. It is a scent we all feel in the whole family, but no one in the family smokes. It’s boring (Patient no. 1).*


Loss of smell and taste could lead to reduced self-esteem; for example, not knowing how salty the food was, leading to a feeling of insecurity or of no longer being able to trust one’s senses when something smelled bad. Smell and taste are essential to many different situations, even to warn of danger; for example, if a fire breaks out.


*I am a teacher in preschool, I work with the small children, I did not sense the smell if they had pooped or whatever, that you had to change the diaper. I did not feel that, so it was disabling in a way (Patient no. 8).*


#### 3.1.2. Lacking Energy

Lacking energy was a symptom experienced for several months after the acute phase of the COVID-19 disease, and patients fought against fatigue to “manage the day”. At the same time, there was a strong desire and drive to be able to do more, although the body and the head were experienced as not being cooperative. The active lifestyle, with exercise and social activities that the patients had had pre-COVID-19, no longer existed, which led to frustration over the impact of lack of energy on their health. There was a strong desire to regain energy and to have the same lifestyle as before COVID-19.


*I think this is probably the worst precisely because the body does not have the strength it wants to have. So I think it has been the worst thing, not being able to go and train at the gym or swim or something like that (Patient no. 12).*


Patients also experienced that social health was negatively affected due to loss of energy. The desire to hang out with friends meant that patients often made appointments but had to cancel on the same day due to lack of energy. Energy levels could vary from day to day but also during the day. It was thus not possible to plan everyday life, and social activities were decided based on their daily condition, which entailed risks of having a bad conscience and the feeling of not being adequate for friends or family.


*It’s the energy, that I do not have the strength, then I do not become this person who has so much tolerance, cares so much about the surroundings, does not have the energy to be as alert at work, does not have the energy to go past my parents after work as I usually do and so on. Of course, it affects health negatively, it is extremely frustrating (Patient no. 10).*


### 3.2. Loss of Control

Loss of control was especially evident when the patients did not recognize themselves because of the symptoms. Everyday life became complicated when the symptoms determined the opportunities that were available. A life of uncertainty was a difficult challenge and created a particular vulnerability. Lack of knowledge regarding long-term symptoms led to anxiety about not being believed, but also a state of indifference, as there were no answers or treatment.

#### 3.2.1. Being Foreign to Oneself

When patients’ physical or mental strength failed, they experienced a feeling of being foreign to themselves. Patients experienced that neither their physical nor mental health was the same as before the COVID-19 disease. There was a feeling of losing control in the face of the symptoms, especially when the ideality and reality became two different worlds. Loss of control, in turn, created feelings of anxiety, a sense of that the situation was unreal in the face of symptoms, and intense frustration.


*The first three months I could not walk up the stairs without feeling shortness of breath, it felt very strange. I was not able to cope with that feeling, it was hard and it took time (Patient no. 1).*


Living with post COVID-19 condition meant pronounced tiredness that manifested itself in all everyday activities. Fatigue created confusion and feelings of discomfort when one’s memory failed. Being foreign to oneself involved a feeling of loss of control, which created a concern for the future and a worry that the symptoms would persist.


*It’s awful, it’s such brain fatigue, like walking in a glass bubble, no one reaches in and no one reaches out in any way. You just want to go to bed. Of course, it’s hard to feel tired and get lost in the thought when patients talk to you, whenever you want to think something intelligent or make a decision (Patient no. 10).*


#### 3.2.2. Seeking Answers

The patients had many questions about the long-term symptoms, but few answers. The uncertainty was based on the patients’ experiences, the environment, and the healthcare system’s ignorance about the symptoms. There was also uncertainty about contacting healthcare services, based on the concern that they would not be taken seriously. There was a striving for answers, where both the health professional and the patients themselves tended to try to find alternative explanations.


*I did not get out of bed. So then I sought medical help or a doctor again, and then I was put on sick leave. And then you were not sure either if it was just the post-COVID symptom that made me tired or if it was some kind of fatigue symptom because the symptoms are quite similar. You consider it a bit like this, which was it, the chicken or the egg (Patient no. 6).*


The patients’ attempts and efforts to seek answers and explanations for the symptoms could eventually turn into indifference, as there were no answers. One way to deal with the uncertainty was to look for alternative explanations for the symptoms in everyday life. Social support was another way of dealing with the uncertainty and subsequent loss of control. Patients experienced relief when they explained their symptoms to relatives, friends, or colleagues. Sharing experiences with loved ones, instead of pretending and keeping feelings to themselves, was important in dealing with uncertainty and regaining control of life.


*Like, to my closest friends, when we sat down and I got to tell how it was and how I felt. Then they got a different understanding that I might not be myself in certain periods. Because… when it is not visible on the outside, it is not always as easy for people to understand (Patient no. 12).*


### 3.3. Revaluation of Life

The patients experienced that post COVID-19 condition caused them to revaluate life, by accepting the transformed body and reflecting on their health and the fragility of life. The acceptance involved in learning to live with the symptoms gave rise to strategies to manage everyday life—partly by accepting that the symptoms existed, but also by listening more to the body and determining what was possible based on their daily condition. Through everyday hard schooling, they learned new insights, such as putting their health first and prioritizing rest and recovery.

#### 3.3.1. Accepting the Transformed Body

The patients described an acceptance of the transformed body as a strategy to continue living their lives, despite the symptoms. Since no answers or solutions were available to cure the symptoms, patients were forced to accept that there were no alternatives but to continue living their everyday lives. There was frustration over the symptoms but also a desire to live despite them, by doing the best they could under the prevailing circumstances. Patients described that their health affected their work, commitments, families, and everyday life. Despite attempts at acceptance, it was also permeated by longing and desire to be the way they had been before the COVID-19 disease.


*It’s not fun. I would like it to be like it used to be. But it’s like I have accepted it, that now it’s just like this. That I can take it easy. It has probably become… that I have to accept it (Patient no. 5).*


When experiencing acceptance, the patients experienced a feeling of listening to the body and accepting the body’s ability, based on the prevailing situation. Sometimes patients experienced weakness in their legs and arms or tremors. Listening to the body and following the body’s signals facilitated and reduced suffering from long-term symptoms. Not being able to do enough for oneself or others, on the other hand, was perceived as suffering. Patients described that they had never listened to their body in that way before, but had gone about at the high pace of everyday life without reflecting on the fact that it could be different. Patients said they had learned to listen to their body and experienced it as something positive. One way to listen to the body could be to find alternative physical activities that better met the body’s new conditions.


*The only positive thing I’ve got out of it is that I have allowed my body to just be, when it needs to rest. And there I don’t feel that I have to perform at my best. But I can swim there in peace and my body feels good about it too. So it has been nice to find… try to find other ways. That it does not have to be the gym or long walks, it can be something simple (Patient no. 12).*


#### 3.3.2. Prioritizing Health

Living with post COVID-19 condition led to a revaluation of what was important in life, such as prioritizing health higher and not taking it for granted, as before. Health was understood to be a crucial element in quality of life. It included physical health, in terms of a healthier and longer life, mental health, such as finding time for recovery, and social health, such as prioritizing the family. Patients described that they were forced to decline social activities more often, which could create a bad conscience, but gradually they felt strong enough to say no in favour of their health. In some cases, while physical health was experienced as being worse than before the COVID-19 disease, mental health was described as stronger than before. Revaluating life also meant realizing that illness can affect anyone. Patients experienced that it became important to talk to loved ones and prioritize their energy to do what was necessary, to live in the present and not worry or dwell on the future or the past.


*I do not focus so much on details anymore, I focus on the life and health for my loved ones. I couldn’t imagine getting sick like this, but it feels like an experience, because I will never again think “that doesn’t happen to me”. It can happen to anyone. You should not be too sure (Patient no. 11).*


Patients experienced gratitude for being alive, and after all, it was possible to live with the symptoms. Gratitude was partly based on patients comparing themselves with others with more severe symptoms. There was an insight into the fact that life does not last forever, which was partly characterized by a previous fear of death. Existential health was given a lot of focus; for example, stories were told about previously having a lot of responsibility at work and assignments, but when the feeling of being mortal came over them, they revaluated their lives. Prioritizing what was important corresponded in part with what patients were grateful for, such as their family, home, beautiful garden, and health.


*I probably feel grateful for life, and I probably did before. But I probably feel that I have reflected even more, and maybe value things even more than I did before (Patient no. 8).*


## 4. Discussion

The results of this study are based on a population of patients with non-severe COVID-19 infection and the small number of patients within this population who had residual symptoms 2 months after COVID-19 infection, i.e., post COVID-19 condition. The results show that the post COVID-19 condition affects the health in this group in a holistic perspective and confirms previous research indicating that even people with mild symptoms can experience long-term symptoms from COVID-19 [4,5]. Our results show that the impairment of health in this group is tied to the fact that everyday life is affected and adapted to the symptoms. The patients’ health experiences moved between uncertainty and new insights. Other studies state similar results, that post COVID-19 condition can contribute to reduced quality of life and impaired performance in daily life [19,31]. Furthermore, long-term symptoms significantly deteriorate health in terms of limitations in cardiorespiratory fitness and muscular strength. These symptoms are severe enough to affect patients’ daily exercise and are not limited to a specific patient or age group [32].

The current study revealed that symptoms were described as lack of energy and loss of abilities, which affected social, existential, physical, and mental health, albeit health was affected to varying degrees. Not being able to live as before and at the same time not knowing what the future would be like created feelings of uncertainty. Health had previously been taken for granted; for example, being able to carry out everyday activities such as gardening, having coffee with friends, or taking exercise—these types of activity could not be performed as before. When patients lose abilities, they change their lifestyle by adapting activities, social events, and habits to their symptoms. Vilhelmsson and Tengland [33] state that health is affected by patients’ own lifestyle choices, but also by determinants that we cannot influence at all, or only to a small extent. The symptoms of post COVID-19 condition were described as a factor over which the patients had no control. The results of the current study show adaptation and new forms of lifestyle, but also the importance of fighting the symptoms and taking control of them, for example going for a walk even though the body is tired. On the other side of fighting fatigue is an increased ability to listen to the body’s needs. The results show increased confidence in assessment of the body’s needs, probably because there is, as yet, no treatment or specific healthcare advice. Crook et al. [34] highlight that fatigue is a common symptom of long-term COVID-19 and the symptoms may overlap with chronic fatigue syndrome. There is as yet no evidence-based management of the symptoms of fatigue, because post COVID-19 condition are still so novel. It has been suggested that rehabilitation interventions may include respiratory, cardiovascular, neuromuscular, and psychological approaches and should be carried out to patients by a multidisciplinary team in order to optimize recovery [35].

The difficult time with post COVID-19 condition was a great challenge, and new insights emerged for the patients in the current study. Despite ill health, the patients looked forward with confidence through a revaluation of life, by accepting the transformed body and prioritizing health. Strategies to create health included, for example, engaging in alternative activities, building hope, and having confidence that the symptoms would go away on their own, but also gratitude that it did not get worse. The results provide knowledge about how post COVID-19 condition influences health, by affecting everyday life to the extent that adjustments need to be made based on health. Other studies show that long-term symptoms of COVID-19 affect work and everyday life [16,18]. The results of the current study show that the feeling that the body is failing and the increased ability to recognize the needs of the body became two sides of the same coin. One side means being forced to adapt because the symptoms become a key part of life. The other side involves lessons for the rest of one’s life, in prioritizing recovery and health. Sun et al. [36] and Zhang et al. [24] lend strength to the assertion that suffering from COVID-19 can lead to a reassessment of life priorities, including a higher appreciation of being alive and a revaluation of values and goals around what is considered important in life.

The current study also shows new insights into life and existential questions about what is important in life, such as time with family and focusing on what creates health for each person and the fragility of life. The results of this study are strengthened by previous research showing that human beings seek and long for meaning and context, which is closely linked to the experience of health, and, when a person suffers from a disease, questions about the meaning of life can arise. The answer to the question “What makes life meaningful?” is highly individual and can change throughout life [37,38]. Also, a holistic perspective includes the importance of the individual experience of health. Bullington and Fagerberg [39] state that a holistic approach encompasses the fact that health is experienced uniquely by each individual.

Findings from the current study shows that the post COVID-19 condition are individually experienced and managed. Furthermore, patients who suffer from post COVID-19 condition are a new group of patients who need to be treated seriously by the health service. This is consistent with previous research showing that a rehabilitation program providing a holistic and multifaceted approach is necessary to care for post COVID-19 condition. Healthcare should strive to individualize treatment and monitor side effects and symptoms, given the current limited knowledge about long-term symptoms [34,40]. More research is needed to investigate the long-term consequences and possible treatment options, to guide patients during recovery from COVID-19 [31,34].

### Methodological Considerations

The qualitative content analysis provides the opportunity to identify and analyze similarities and differences in a phenomenon. It does not strive to identify consensus, but rather to deepen knowledge and understanding [25,28]. Post COVID-19 condition constitutes a new phenomenon. The method is considered suitable to gaining increased knowledge of how individuals understand and experience their health after being affected by post COVID-19 condition.

In qualitative research, trustworthiness is defined according to the four criteria: credibility, dependability, confirmability, and transferability [25,28]. *Credibility* refers to confidence in the truth of data and analysis [25,28] and was strengthened in this study, because the data were coded and analyzed until consensus was reached in the research group. The results are based on the participants’ experiences and strengthened by quotations from the participants, but no member check was performed. *Dependability* refers to the stability of data over time [25,28]. In this study, it was strengthened in that the interviews were based on an interview guide, thus asking the same questions to all participants, tested in two interviews, and conducted by an experienced qualitative researcher (IL). The authors’ various professions—in social work, nursing, and medicine—and various academic experiences—professors, associate professors, doctor, doctoral student, and master’s degree—as well as various gender—women and men—meant that the analysis was based on different perspectives, which counteracted the fact that preconceptions limited the study. *Confirmability* refers to the neutrality of data [25,28] and was strengthened by the methodological process that was carried out throughout the analysis process, in which all steps in the analysis have been reported and described carefully. A limitation could be the small number of patients, but the interviews were assessed to be rich and contained a wide variety of content. The richness is manifest in that each sub-category is represented by content from between five to ten interviewees. The result shows saturation since no new sub-categories or categories emerged after the seventh interview. *Transferability* refers to whether the results apply to groups other than the participants in the study [25,28]. This study included men and women of different ages and living conditions, which increases the transferability of the study. The study aimed to be transparent and thoroughly executed, which increases transferability. Limitations may be that data were collected from the same geographical region and all participants were born in Sweden.

## 5. Conclusions and Implications

Patients’ health experiences of post COVID-19 condition moved between uncertainty and new insights. Loss of abilities and loss of control led to feelings of uncertainty and worries about the future. Uncertainty was expressed as an inability to trust some senses, such as smell and taste, but also in relation to the absence of answers about the post COVID-19 condition. There was a fear of contacting the healthcare service because they might not be taken seriously, but also a fear that there were no answers to be found.

The symptoms influenced everyday life, with lack of energy affecting work performance and social life. The patient’s daily condition determined their energy levels, and social activities were often cancelled on the same day. Their physical health was affected by the limited opportunities for training activities because their cardiorespiratory fitness and muscular strength was not the same as before COVID-19. Despite experiencing ill health, patients looked forward with confidence by revaluating life, including accepting their transformed body and prioritizing health. The difficult time with long-term symptoms was challenging, and new insights emerged, such as being grateful to be alive, but also insights about new priorities, such as time with the family and rest when the body needed it.

This study shows that patients with post COVID-19 condition need to be treated flexibly, based on each patient’s unique challenges, how they experience symptoms and their consequences for their health. Further research is needed to study the salutogenic factors and reduce the burden that the post COVID-19 condition have on health. Further research on how society can improve information and nursing in this context is recommended, with the aim of increasing health for this patient group. Although in-depth research is also needed on specific symptoms, such as lack of energy and impact on patients’ lives, what is principally required is more knowledge on how society can promote the health of affected patients.

## Figures and Tables

**Table 1 ijerph-19-13980-t001:** Sociodemographic data of the 14 patients.

Variable	*n*
**Sex**	
Women	10
Men	4
**Age in years,** Median (range)	58 (26–76)
**Levels of education**	
Secondary school	7
University	7
**Civil status/**	
Co-habiting	12
Living alone	2
**Employment**	
Working full time	5
Working part-time (25–80%)	6
Retired	3
**Duration of post COVID-19 condition**	
5–6 months	2
16–21 months	12
**Most frequent long-term symptoms**	
Anosmia/taste disorder	7
Fatigue	7
Dyspnoea	5
Impaired physical condition	3

**Table 2 ijerph-19-13980-t002:** Overview of the theme, categories, and sub-categories constructed from the qualitative content analysis of patients’ health experiences of post COVID-19 condition.

Theme	Moving Between Uncertainty and New Insights		
**Categories**	Loss of abilities	Loss of control	Revaluation of life
**Sub-categories**	Losing smell and taste	Being foreign to oneself	Accepting the transformed body
	Lacking energy	Seeking answers	Prioritizing health

## Data Availability

Not applicable. The data will not be shared, because the ethics approval for the study requires that the transcribed interviews be kept in locked files, accessible only to the researchers.

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
