# Peer review of "Patients’ Health Experiences of Post COVID-19 Condition—A Qualitative Study"

_ijerph, 2022, doi:10.3390/ijerph192113980_

Round 1

Reviewer 1 Report

Thank you for the opportunity to review this qualitative study on LongCovid, a subject which, although 2,5 years into the pandemic, has not been extensively investigated.

My main concern regards the possble phenotypes of the interviewed subjects. Do you have any specific diagnosis such as for example autonomic dysfunction, respiratory issues etc? Can you describe them more specifically for eaxh patient?

Depending on the underlying phenotype and symptom clusters patients can be differently affected which also influences both their experiences with loss of abilities as well as coping mechanisms.

Please also revise the English language, preferably by someone proficient in English. The term""reevaluate" is mainly written as "revaluate" which is not correct. Please revise throughout,

Reviewer 2 Report

Almgren et al. present their paper "Patients' health experiences of post COVID-19 condition - a qualitative study" on patients suffering from long-term symptoms.

The study included a total of 14 patients from an ongoing trial of COVID-19 who were affected by symptoms that lasted longer than 8 weeks. Data were collected through individual semi-structured interviews. Patients' qualitative health experiences after COVID-19 varied between uncertainty and reassessment of life, including prioritization of health. The study is well structured, very interesting and offers new insight into the reality of the lives of those affected as well as the resulting conclusions.

However, the study has some limitations. Only a very limited number of patients was included in the study. At least in the introduction, I would like to see more references to the frequency of long or post COVID-19 conditions. This could then be taken up again in the discussion in order to put the very small number of patients with post COVID-19 from the mentioned study with 360 patients, from which these were recruited, into relation again.

In the introduction it was also mentioned that even patients with mild initial symptoms can experience long-term symptoms. This should be discussed in more detail again for the recruitment of the patients, who are "non-hospitalised patients", with regard to the severity of the possible post COVID-19 symptoms.

Even though the paper can provide rather anecdotal insights due to the small number of patients, it does list in the chapter “Methodological Considerations” the most important criteria in qualitative research: credibility, dependability, confirmability, and transferability. The discussion of the paper here is also clear and appropriate. Despite the partly anecdotal character, the response frequencies of the categories listed in Table 1 should be noted if possible (if not already done, at least as a supplement). Even though this is a qualitative paper, some impression of the distribution of responses would be interesting for the readership.

The study is relevant and useful for the medical practitioner who will be dealing with patients affected by long or post Covid-19 with increasing frequency. Therefore, the work should be published after the above points have been completed.

The following minor comments should also be taken into account:

-       In the introduction, the different definitions of long and post COVID-19 should be presented in more detail (WHO definition etc.) and the time frames considered (2 months / 3 months) should be explicitly presented again in terms of onset and duration

-       Readers would certainly be interested in a brief overview, based on appropriate references, of how often the various symptoms/symptom complexes occur in long or post COVID-19 (e.g. "brain fog", fatigue, ...)

-       Readers may also be interested in the discussion of other approaches to obtaining data on the reality of life, e.g., using search engine data, as shown in the example of the representation of long COVID-19 (e.g., Kaatz et al. "Representation of long COVID syndrome in the awareness of the population is revealed by Google Trends analysis." Brain, Behavior, & Immunity-Health 22 (2022): 100455), which could be included in the discussion

Reviewer 3 Report

In the study, qualitative information was obtained about the experiences of patients who showed symptoms for more than 2 weeks after COVID-19. The choice of topic is far from original, since hundreds of studies have been conducted about COVID-19 for more than 2 years. However, it can benefit from a holistic treatment approach to the disease, guide future studies in terms of COVID-19, and study data can contribute to the literature. The article needs major and minor revision before publication. My Suggestions:

Major

1. According to COREQ criteria; please mention the researcher's profession / academic title, gender, professional experience and education. (Domain 1)

2. According to COREQ criteria; Has a pilot study been conducted? Were the interviews repeated and how many repetitions were made? What is the interview time per person? Were transcripts forwarded to the participants so that they could comment/confirm? (Domain 2 Data collection)

3. According to COREQ criteria; If software is used in data management, specify which one is used (Domain 3 Analysis)

Minor

1. Writing the open state of SARS-CoV-2 when it is first mentioned in the introduction section.

2. In the same way, PCR should also be written in parentheses.

Round 2

Reviewer 3 Report

Thank you for revision.